# Study on Adsorption Behavior of Nickel Ions Using Silica-Based Sandwich Layered Zirconium-Titanium Phosphate Prepared by Layer-by-Layer Grafting Method

**DOI:** 10.3390/nano11092314

**Published:** 2021-09-06

**Authors:** Chunmin Li, Jinsheng Zhao, Yusheng Zhang

**Affiliations:** 1College of Chemistry and Chemical Engineering, Liaocheng University, Liaocheng 252000, China; zhaojinsheng@lcu.edu.cn; 2CAS Key Laboratory of Nuclear Materials and Safety Assessment, Institute of Metal Research, Chinese Academy of Sciences, Shenyang 110016, China; yshzhang19b@imr.ac.cn

**Keywords:** silica substrate, Ni adsorption, zirconium-titanium phosphate, thermodynamic simulations

## Abstract

In this study, the composite of silica-based sandwich-layered zirconium-titanium phosphate was prepared by a layer-by-layer grafting method and its adsorption properties in a diluted solution of Ni ions were specifically researched by the bath experiment method. The field-emission scanning electron microscope (FESEM) results presented the smooth surface morphology of the pristine adsorbent and a rough surface morphology of the adsorbed adsorbent and the energy dispersive analysis (EDS) results ensured the presence of the original metal element (Si, O, Ti, P, Zr) and the captured nickel element on the adsorbent. The Fourier transformed infrared spectroscopy (FTIR) revealed the new band formation of -Si-Ti-O-, -Si-Ti-O-P-, and -Si-Ti-O-P-Zr-O-, which ensured the successful modification of the silica substrate by zirconium-titanium phosphate. The specific surface area and pore size distribution analysis indicated that the pore structure was changed from type-Ⅳ to H2-type and the specific surface area (BET) of the modified composite was 337.881 m^2^/g. In the bath experiment, the optimal pH for adsorbing Ni ions on the composite was ~8 with the equilibrium time 30 min at room temperature and the maximum sorption amount was 50.1 mg/g. The adsorption kinetics of the sorption process were corresponded to the pseudo-second-order kinetic equation and the isothermal adsorption data were fitted well to the Redlich-Peterson Model. Thermodynamic simulation results revealed the species of Ni ions and provided a reasonable pH scope for better removal of the Ni element in wastewater.

## 1. Introduction

Heavy-metal ions at dilute concentrations widely exist in the environment, mainly from the drainage of industrial sewage. Dilute concentration heavy-metal ions, namely, the content of heavy-metal ions in the industrial wastewater are below ppm grade. This concentration can cause permanent harm to human body and the environment ecology because any physical and chemical method or microbial processing cannot degrade heavy-metal ions and just change its existent form [1]. Although the content of heavy-metal in some water bodies is within a certain safe range (the concentrations of total Ni (TNi) must be less than 0.1 mg L^−1^ in China [2]), it could accumulate in plants and animals in liquid phase and cause permanent damage. Wastewater containing low-level heavy Ni^2+^ has a great and far-reaching impact on the environment and has caused indelible harm to plants and animals [3,4]. Therefore, it is strategically and ecologically urgent to accelerate research on the treatment of low-concentration Ni^2+^ in solution.

The sorption processes are well known to be an efficient way to remove heavy metals particularly in these dilute systems, which provide the most important processes controlling contaminant mobility in the environment [5]. Many articles using organic or inorganic adsorbents have been reported, such as metal-organic frameworks [6,7,8], clays [9], zeolites [10], birnessite [11], modified activated carbon [12], gallocyanine-grafted hydro-gel [13], nanoporous polymers [14], Fe_3_O_4_ particles [15], Ni_0.8_K_0.2_Fe_2_O_4_ nanomaterials [16], and so on. However, corrosion and mechanical stability problems are normally found in the organic adsorbents while narrow pH applications and low ion selectivity often exist in metal oxide adsorbent [17]. Furthermore, besides the surface properties of the adsorbent and the environment of the aqueous solution, most relevant retention mechanism studies did not take the existent form of heavy-metal ions into consideration as the metal nature which includes complex cations, free cations, free oxyanions [18], or the formation of precipitates at the interface as hydroxides or insoluble oxides [19], which is one of the most important aspects affecting the removal style. It is well known that trace-level adsorption (ion-exchange) is often completely different compared with macro-level processes. Therefore, thoroughly exploring the key controlling parameters and the retention mechanism in more dilute systems could be favorable to develop accurate sorption models which can effectively predict the transport and fate of the environmental heavy-metal ions and achieve a more in-depth understanding of sorption process.

Tetravalent metal phosphate with the general formula M(IV)(HPO_4_)_2_·nH_2_O, where M(IV) could be Zr, Ti, Sn, Ce, Th, etc., possess great properties in the applications of adsorption, ion exchange [20], catalysis [21], proton conduction [22], and so on. For example, titanium phosphate (TiP) and zirconium phosphate (ZrP) with excellent adsorption amounts, abundant -P-OH or -OH group active sites, and special layered structures with a hexagonal shape are widely known as suitable adsorbents or ion exchangers to remove heavy metal ions from aqueous solutions [23]. Although many specially designed structures of metal phosphate for removing target metal ions have been developed [24,25,26,27], their superiorities are constrained by small surface areas, small pores, and interlayer spaces caused by the cluster formation. Besides, tetravalent metal phosphates containing single-metal cation are inferior to those compounds containing two different anions and a cation in aspects of sorption or exchange properties and selectivity for target hazard ions [20]. However, little work has been carried out to research metal phosphates with two cations, which may be more advantageous to remove specific hazard ions. Although, some articles have reported zirconium titanium phosphate and evidenced better potential for this material to be used as a cation exchanger than monometallic phosphate, reporting the replacement of zirconium with titanium or titanium with zirconium in phosphate structure [28]. Hence, the morphological control of a Ti-Zr adsorbent with the assistance of reliable support may enhance the adsorption performance of adsorbent [2,29]. To the best to our knowledge, there are hardly any articles reporting the preparation of polymetallic phosphate-containing multilayer structure with each layer containing different structural metal atoms. Based on the abovementioned, loading polymetallic phosphate one layer at a time onto suitable carriers at the molecular level to form composite adsorbents or ion exchangers will be a very promising prospect in the disposal of heavy-metal ions.

SBA-15 is a kind of mesoporous molecular sieve. It has wide application prospects in the fields of catalysis, separation, biology, and nano materials. Based on its advantages of high hydrothermal stability, large surface area, and abundant surface hydroxyl groups, it has opened up a new research field for catalysis, adsorption separation, and advanced inorganic materials. Besides, it was also used as a support to make the composites more stable [30]. In this work, SBA-15 was chosen as the silica-based substrate according to its abundant -OH modified sites. The developed layer-by-layer method was adopted, which could allow for a better control of the alternating layers of zirconium titanium phosphate on SBA-15 at the molecular level and form high-k dielectric materials with lower porosity [31]. The rationale behind our choice of precursors for surface modification is based on the fact that Ti(OPri)_4_, Zr(OPri)_4_ and POCl_3_ can be used as suitable “acid-based” pair precursors for the synthesis of zirconium titanium phosphate [30]. The structure of the synthesized composites, the adsorption behavior of heavy metal ions, and the adsorption mechanism were studied in detail in this work. The novel silica-based zirconium titanium phosphate adsorbent with alternating layers showed good application prospects in the low-concentration heavy-metal ions removing process.

## 2. Material and Methods

### 2.1. Materials

The SBA-15 was synthesized according to reference [31]. Analytical grade zirconium propoxide was obtained from Shanghai Macklin Biochemical Co., Ltd. (Shanghai, China; Purity: 70 ωt% in 1-propanol). Analytical grade titanium isopropoxide (Beijing, China; Purity: 99.9% metals basis) was purchased from Alfa Aesar Co., Ltd. Analytical grade phosphorous oxide trichloride (Purity: 95%) was purchased from Sinopharm Group Chemical Reagent Co., Ltd. (Guangzhou, China). Analytical grade KCl (Purity: 95 ωt%), MgCl_2_ (Purity: 99.5 ωt%), CaCl_2_ (Purity: 96 ωt%), Ni(NO_3_)_2_ (Purity: 98 ωt%), HCl (Purity: 38 ωt%) and NaOH (Purity: 98 ωt%) were all obtained from Guangzhou Jinhua Da Chemical Reagent Co., Ltd. (Guangzhou, China). MilliQ water was used throughout the experiments (electric conductivity: 0.055 us/cm).

### 2.2. Material Synthesis Process

As shown in Figure 1, the synthetic procedure of the adsorbent was as follows. Step 1: The SBA-15 silica-based substrate was impregnated for 20 min at 80 °C in a piranha solution (a mixture of concentrated sulfuric acid and 30% hydrogen peroxide (7:3)) in order to clean the surface and generate a well-producible oxide layer. After the treatment, SBA-15 was filtered by suction filtration and washed four times by water. Step 2: 1 g of the pretreatment SBA-15 was added into a three-neck flask with two glass stoppers and 20 mL of anhydrous toluene and 3.4 mmoL titanium isopropoxide were added into the flask. Step 3: After sealing with two glass stoppers and a condenser pipe, the flask was refluxed with vigorous stirring under an oil bath for 2 h at 110 °C. The filtered sample noted as Si-Ti was washed three times by anhydrous toluene and three times with deionized water for hydrolysis. The sample was dried in a drying oven at 80 °C for 24 h. Step 4: The dried sample Si-Ti was then reacted with 10.2 mmoL phosphorus oxychloride by the same procedure above and the reacted production was noted as Si-Ti-P. Step 5: The sample noted as Si-Ti-P finally reacted with 3.4 mmoL zirconium propoxide solution by repeating the same procedure and the final adsorbent was noted as Si-Ti-P-Zr.

### 2.3. Characterization Methods

Fourier-transformed infrared spectroscopy (FTIR) of the samples was recorded from 400–4000 cm^−1^ to evidence the formation of the main functional groups before adsorption and the new metal-chemical bond after adsorption (Ramsey, MN, USA). The crystalline phase of the samples was analyzed by X-ray diffraction instrument (XRD), in which Ni-filtered Cu-Kα radiation were analyzed with a scanning rate of 8°/min from 10 to 60° and a step size of 0.02° (SMARTLAB3KW, Tokyo, Japan). The morphology analysis of the adsorbent before and after adsorption was examined by the field-emission scanning electron microscope (FE-SEM), (Hitachi SU8220, Tokyo, Japan). The elemental content of the adsorbent before and after adsorption was analyzed using energy dispersive analysis (EDS, EDAX PV8200, Tokyo, Japan). The lattice structure of the sample was analyzed by Spectra S/TEM type transmission electron microscopy (TEM) (Spectra S, Amsterdam, Netherlands). The sample thickness was less than 1 mm and the acceleration voltage was 300 kV. The specific surface area (BET) and pore distribution of Si-Ti-P, Si-Ti-P, and Si-Ti-P-Zr were measured by the surface area and porosity instrument with N_2_ adsorption and desorption methods (Micromeritics TriStar II, New York, NY, USA).

### 2.4. Ni^2+^ Adsorption Experiment on Zr-P-Ti-Si in Batch Experiment

Typical adsorption experiments of the Ni ions were conducted as follows: 0.5 g of the nitrate of heavy metal ions was dissolved into 500 mL ultrapure water, forming 500 ppm stock solution. The diluted solutions were obtained by adjusting the stock solution to the following experiments requiring different concentrations of heavy metal solutions; 0.05 g of the silica-based zirconium titanium phosphate powder was added into 50 mL of 500 ppm Ni ions solution in the polyethylene tube, and then the capped tub was shaken at table concentrator with 100 rpm for 24 h at room temperature to reach equilibrium. The adsorbed material was separated by a centrifuge at 8000 rpm and respectively washed three times by pure water and ethyl alcohol, then dried in a drying oven at 80 °C. The supernatant was again filtered by drainage pin-type filter (0.45-μm diameter) and the concentration of the metal ions in the filtered solution was measured using Graphite Furnace Atomic Absorption Spectrometry (GFAAS), (AA-700, Shimadzu, Kyoto, Japan) for low concentrations (ppb grade). The adsorption capacity, the distribution coefficient, and the removal efficiency were calculated by the following Equations [32,33]:(1)$$q=(V\left[C_{0}-C_{F}\right])/m$$
(2)$$K_{d}=V\left[\frac{\left(C_{0}-C_{f}\right)}{C_{f}}\right]/m$$
(3)$$R\%=100\times (C_{0}-C_{f})/C_{0}$$
where q (mg/g) is the adsorption capacity of the silica-based zirconium titanium phosphate; $K_{d}$ (mL/g), used for the determination of the affinity and selectivity of the compounds for different metal ions; R% represents the removal efficiency; $C_{0}$ (mg/L) is the initial concentration of the heavy metal ions; $C_{f}$ (mg/L) is the residual concentration of the metal ions in solution.

#### 2.4.1. The Effect of the Initial Concentration of Ni Ions on the Sorption Process

The study on the various concentrations (0–250 μg/L) of Ni ions was carried out using batch method with a solid/liquid ratio of 1 g/L (0.1 g/0.1 L), contact time 24 h, pH = 8, and at room temperature. The residual concentration of Ni ions was filtered and measured by GFAAS.

#### 2.4.2. The Effect of Contact Time on the Sorption Process

The kinetic study was conducted by dispersing 0.1 g of the adsorbent into 100 mL solutions with the heavy metal concentrations of 60 μg/L and the contact time range from 0–120 min (sampling time point: 2, 5, 10, 20, 30, 60, 90, 120 min). The pH of the solution was 8 and the adsorption process was conducted at room temperature. The suspensions were filtered at the various reaction times, and the filtrates were analyzed by GFAAS.

#### 2.4.3. The Effect of pH on the Sorption Process

The effect of pH on the adsorption process was measured by dispersing 0.1g of the adsorbent into 100 mL solutions with the heavy metal concentrations of 60 μg/L, and the pH scope of the solution ranged from 1 to 14 and was adjusted by 1M HCl and 1M NaOH solution and sodium hydroxide solution; then, the capped tubes were shaken at 100 rpm for 3 h to reach adsorption equilibrium at room temperature. All the suspensions were filtered and analyzed by GFAAS.

#### 2.4.4. Thermodynamic Simulations of the Hydrolyzed Species of Ni Ions

The solution pH caused the hydrolyzed species and the complete precipitation of the heavy metal ions, which affects the adsorption process on the adsorbent. To understand the adsorption behavior of heavy metal ions on the adsorbent, thermodynamic simulations were utilized to reveal the existent form of heavy metal ions in various pH (1–14) solutions. The thermodynamic calculations were carried out with Matlab^®^R2010a software using thermodynamic parameters of all possible reactions to simulate the possible species of heavy metal ions including metals ions and metal ion complexes. Equations (4)–(7) were used to calculate the species distribution of Ni ions at different pH values. These simulation results together with experimental results facilitate the understanding of adsorption mechanism. The adsorption experiments were conducted at specific pH values which possess the specific hydrolyzed species of Ni ions. The adsorbed material was separated by filtration and analyzed.

Chemical reaction of Nickel hydroxide species at different pH values (all the data were obtained from HSC 6.0 database):(4)$${\mathrm{Ni}}^{2+}+{\mathrm{OH}}^{-}\to {\mathrm{NiOH}}^{+}\;\mathrm{Ksp}=4.97$$
(5)$${\mathrm{Ni}}^{2+}+2{\mathrm{OH}}^{-}\to \mathrm{Ni}{\left(\mathrm{OH}\right)}_{2}\;\mathrm{Ksp}=8.55$$
(6)$${\mathrm{Ni}}^{2+}+2{\mathrm{OH}}^{-}\to \mathrm{Ni}{\left(\mathrm{OH}\right)}_{2}\left(s\right)\;\mathrm{Ksp}=16.96$$
(7)$$Ni^{2+}+3OH^{-}\to Ni{\left(OH\right)}^{3-}\;\mathrm{Ksp}=11.33$$

#### 2.4.5. The Affinity Measurement toward Ni Ions

To measure the selectivity property toward Ni ions of the adsorbent, the counter ions such as K, Mg, and Ca, which largely existed in wastewater and make strong competitors for the adsorption sites in many adsorbents, were selected. The distribution coefficient $K_{d}$ was used for test the affinity of the adsorbent for Ni ions and the equation is given in Equation (3).

The selectivity test was conducted using mixed solutions with 50 mL 0.5 mmol L^−1^ of K, Mg, and Ca ions and 60 ug L^−1^ of Ni ions at room temperature and the pH of the solution was adjusted to about 8 using 1 M NaOH solution. The dosage of the adsorbent was 0.06 mg and the contact time was 2 h to ensure sorption equilibrium.

## 3. Results and Discussion

### 3.1. Characterization Analysis

#### 3.1.1. TEM Analysis of the SBA-15 and SiO_2_-Ti-P-Zr Adsorbent

In order to characterize the morphology of the composite, TEM characterization of SBA-15 support and final SiO_2_-Ti-P-Zr adsorbent was illustrated in Figure 2. From Figure 2a, we can see that the SBA-15 support processed no layer structure before loading the functional materials, which corresponds to the magnified image in Figure 2a-1. Compared with SBA-15 support, Figure 2b, which represents the cross-section image of the adsorbent, shows the layer-by-layer shape and there is a clear boundary between functional materials and SBA-15 support shown in Figure 2b-1. Specific interlayer width is described in Figure 2c. From Figure 2c-1, we can see that the width of -Ti-O-P- layer was about 11.38 nm while the width of -P-Zr-O- layer was about 7.25 nm. Therefore, the width of one constitutional unit of the adsorbent was 18.63 nm. EDS analysis indicated that molar ratio of the element (Ti: O: P: Zr) on the surface of the adsorbent was 9.14: 59.15: 15.44: 1.54. According to the measurement and calculation of the nanoscale structure in the HRTEM image, the Si-Ti-P-Zr lamellar structure effectively grew on the Si- based support.

#### 3.1.2. The FT-IR and XRD Analysis of the Formation Process of the Adsorbent

The FT-IR analysis of the formation process of the adsorbent including SiO_2_, SiO_2_-Ti, SiO_2_-Ti-P, and SiO_2_-Ti-P-Zr is illustrated in Figure 3a. The peaks at 3446 and 1663 cm^−1^ respectively corresponded to the antisymmetric and symmetric stretching vibrations of -OH groups in water [34]. The peak at 1090 cm^−1^ was attributed to the representation of antisymmetric vibration of Si-O-Si [35], and peaked at around 793, 966, and 470 cm^−1^, responding to symmetrical stretching vibration of Si-O [35]. As elucidated in Figure 3a, the peak at 793 cm^−1^ gradually disappeared as the synthesis process progressed from Si-Ti to Si-Ti-P-Zr, which indicated that the Si-O bonds acted as active sites bonding to the raw materials in the synthesis process. Based on previous report, the peak of O=P band should be around 1006 cm^−1^ [36,37] which caused a wider and larger peak around 1090 cm^−1^ in Si-Ti-P, indicating that the peak of O-P was merged with the peak of Si-O-Si. These bands indicate the presence of structural hydroxyl groups/protonic sites in the material.

The schematic diagram of the XRD of the adsorbent during the synthesis process is described in Figure 3b. No diffraction peaks were obtained either in the intermediate product or the final product, which indicated the amorphous structure existed during the whole synthesis process. Based on our previous study [38], adsorbent with amorphous structure exhibited a satisfactory metal adsorption. Herein, the amorphous structure of the adsorbent may be a significant factor for the better adsorption property of Ni ions.

#### 3.1.3. The Specific Surface Area and Pore Size Distribution Analysis

The isotherm curve obtained from adsorption-desorption of nitrogen and the pore volume were performed to identify the hole type of the modified porous materials, which are described in Figure 4. The result indicates that the BET and pore volume gradually decreased in the modification process, while the average pore diameter hardly changed in the process. The BJH model was used to analyze the desorption curve of the different materials. As shown in Figure 4a–d, four kinds of materials exhibited a representative type-Ⅳ adsorption behavior, which illustrates a strong force between the material and nitrogen. In Figure 4a, a narrow H1-type hysteresis loop appeared at a relative pressure between 0.7 and 1 due to the capillary coagulation. This indicated the sample mesoporous was ordered. In Figure 4b–d, the modified samples are shown to have had a broad H2-type at a relative pressure between 0.4 and 1, which was attributed to the porous adsorption and uniform particle packing hole. The transformation of the hysteresis loop type reveals the existence of mesoporosity and macroporosity. Based on the above analysis, the BET and pore structure of the adsorbent were greatly altered in the modification process. The BET, pore volume, and average pore width data obtained from three fresh samples are listed in Table 1. As shown in Table 1, the BET specific surface area significantly reduced from 566.991 to 337.881 m^2^/g, which indicated that the adsorbent effectively grew on the surface of the silica support and reduced the surface area. Additionally, the formation of the adsorbent changed the pore structure of the SBA-15 from type-Ⅳ to H2-type. Besides, the pore volume and the average pore width also reduced from 70.787 Å to 66.997 Å through the synthesis process.

#### 3.1.4. The Morphology and Element Mapping Analysis of the Pristine Adsorbent and Adsorbed Adsorbent

The morphology analysis of the pristine adsorbent and adsorbed adsorbent was conducted by FESEM and the results are shown in Figure 5. As presented in Figure 5a, the morphology of the pristine adsorbent was smooth and neat while the adsorbent after adsorbing Ni^2+^ showed a rough morphology which was caused by the chemical bond formation of -P-O-Ni or -O-Ni on the surface of the adsorbent which was corresponded to the results of FT-IR. The element distribution of the pristine adsorbent and adsorbed adsorbent are shown in Figure 6. Element mapping analysis of the pristine adsorbent and adsorbed adsorbent ensured the presence of the original metal element (Si, O, Ti, P, Zr) and the captured nickel element. Besides, all the metal elements were homogeneously distributed on the surface of silica substrate. Figure 7 shows the atom percent of the elements. Those results indicated that O was the dominating element (74.84%) which provided abundant hydroxyl adsorption sites for adsorbing Ni ions. The atom percent of the captured nickel element was 0.33%, which revealed the relative content of Ni atoms. The reason for the lower content of Ni atoms may be ascribed to the very low concentration of Ni solutions used throughout the experiment.

### 3.2. Analysis of the Hydrolyzed Species of Ni Ions by Thermodynamic Simulations Method

Figure 8 presents the results of the thermodynamic simulations of the hydrolyzed species of Ni ions in the pH value between 1 and 14. In the pH region from 1 to 8, the occupied Ni species was Ni^2+^. When the pH > 8, Ni^2+^ began to reduce and the Ni species of NiOH^+^ and Ni (OH)_2_(aq) started to appear. When the pH was about 10, the occupied Ni species was Ni (OH)_2_(aq) while Ni (OH)^3-^ began to appear and disappeared when the pH was >13. The Ni species started to appear when the pH was about 9 and always stayed in the solution until the pH was 14. These results indicated that the pH of the solution had a great influence on the hydrolyzed species of Ni ions, while the concentration of hydrolyzed Ni ions with more than two hydroxyls was extremely low based on the low Ksp values (8.55 for Ni (OH)_2_ (aq), 16.96 for Ni (OH)_2_ (s) and 11.33 for Ni (OH)^3−^). Therefore, the appropriate pH for studying the sorption of Ni ranged from 1 to 10 as most Ni ions may precipitate when the pH > 10.

### 3.3. Analysis of the Batch Adsorption and Ion Selectivity

The results obtained from the effect of the initial concentration on the adsorption of Ni ions on the adsorbent are shown in Figure 9a. The adsorbed amount increased with the increasing of the initial concentration of Ni ions. When the initial concentration of Ni ions exceeded 52 μg/L, the adsorption achieved equilibrium and the largest sorption amount was 50.1 mg/g. Therefore, 52 μg/L of Ni ions was selected as the best initial concentration in the following experiment.

Figure 9b shows the influence of the pH on the sorption amount of Ni. From Figure 9b, we can conclude that the sorption amount increased with the increasing pH from 1–6 and it maintained a slight change at the maximal sorption amount (50.1 mg/g) in the pH range from 6 to 10. When the pH was >10, the sorption amount rapidly decreased due to the low concentration of Ni species in the solution, which the Ni species transformed from NiOH^+^ to Ni (OH)_2_ precipitate (Figure 8). Besides, the low sorption amount in low pH region may be attributed to the reason that excessive hydrogen ions occupy the sorption sits on the adsorbent. Therefore, the pH ~8 was selected as the optimal one in this study.

The influence of contact time is described in Figure 9c. Figure 9c shows that the sorption equilibrium was achieved within 30 min, which presented a rather rapid adsorption ratio of Ni ions on the adsorbent. The fitting curve with the second-order dynamic model is illustrated in Figure 9c. It evidenced that the sorption process satisfied the second-order dynamic equation well and the fitted straight line.

The affinity and selectivity test results from multicomponent solutions are shown in Figure 9d. The $K_{d}$ values of the adsorbent toward K, Mg, Ca, and Ni in these experimental conditions were found to be 9.4, 153.8, 161.2, and 3501.3 mL/g respectively, which indicated a better affinity of the adsorbent toward Ni ions and the order was Ni^2+^ (9.4 mL/g), Ca^2+^ (161.2 mL/g), Mg^2+^ (153.8 mL/g), K^+^ (9.4 mL/g). Based on our knowledge [39], the hydrate species and valence state of metal ions in aqueous solution were also influenced by the sorption process on the adsorbent, as some metal ions are commonly removed from aqueous solutions by coprecipitation. As can be seen in Figure 9d, the low $K_{d}$ value of K can be mainly attributed to the low valence state (monovalence), while the relatively higher $K_{d}$ values of Ca and Mg are due to the higher valence state (divalent). The difference of the $K_{d}$ value between Ni and Ca, Mg may be ascribed to the hydrated radius of Ni^2+^ being more appropriate than that of Ca^2+^, Mg^2+^ when considering the interlayer size of the adsorbent.

### 3.4. Adsorption Data Simulation by Isothermal Adsorption Models

To better understand the sorption process in dilute aqueous solution containing Ni ions, isothermal adsorption models such as Langmuir, Freundlich, and Redlich-Peterson isotherms sorption models were introduced to fit the sorption data and the equations are given as follows:

Langmuir Model [40,41,42]:(8)$$q_{e}=\frac{q_{m}K_{L}C_{e}}{1+K_{L}C_{e}}$$
where $q_{e}$ (mg/g) is the equilibrium absorption capacity; $C_{e}$ is the equilibrium concentration of Ni ions in solution (mg/L); $q_{m}$ (mg/g) is the estimated saturated adsorbed amount of the adsorbent; $K_{L}$ (L/mg) is the isothermal adsorption equation constant.

Freundlich Model [43]:(9)$$q_{e}=K_{F}C_{e}^{1/n}$$
where $q_{e}$ (mg/g) is the equilibrium absorption capacity; $C_{e}$ is the equilibrium concentration of Ni ions in solution (mg/L); $K_{F}$ is the sorption parameter, namely the distribution coefficient of nickel ions between aqueous phase and adsorbent; n is the constant.

Redlich-Peterson Model [44]:(10)$$q_{e}=AC_{e}/\left(1+BC_{e}^{g}\right)$$
where $q_{e}$ (mg/g) is the equilibrium absorption capacity, $C_{e}$ is the equilibrium concentration of Ni ions in solution (mg/L); A, B are the Langmuir-Freundlich parameters and g is the Freundlich empirical parameter, which is between 0 and 1.

The nonlinear fitting results with those models are shown in Figure 10. The calculated equilibrium constants are summarized in Table 2. The correlation coefficient (*R*^2^) suggested that the Ni adsorption on the adsorbent could be well described by the Redlich-Peterson isotherms adsorption model, which may explain monolayer uniform adsorption with finite equivalent adsorption sites and multilayer adsorption on the heterogeneous surfaces which corresponded to the results in Figure 2b. The lower values of the correlation coefficient in the Langmuir and Freundich models evidenced that the Redlich-Peterson isotherms adsorption model was more suitable when used for describing the actual situation of solution adsorption. However, according to the complexity of the solid-liquid sorption, it should be noted that the Redlich-Peterson isotherms adsorption model could not completely describe the reaction mechanism of the complex system, where it can just be used as a method for processing experimental data.

### 3.5. Ni Ions Leaching and the Stability of the Adsorbent Test

The leaching experiment was conducted by putting 0.1 g of the adsorbed adsorbent into 50 mL of 0.1 M HCl solution at room temperature and the leaching time was from 1 to 14 days. The leaching amount of Ni ions was measured by GFAAS and the results are shown in Figure 9. In addition, the leaching rate (*L*) was obtained by the following Equation [44,45]:(11)$$L=\frac{C_{t}*V}{q_{e}*m*1000}$$
where *q_e_* (mg/g) is the equilibrium absorption capacity; *m* is the mass of the adsorbent, *C_t_* (mg/L) is the Ni concentration of the leachates at time *t*; and *V* (mL) is the volume of the leaching solution.

From Figure 11, we can see that the leaching ratio of Ni slightly increased with the contact time from 1 to 14 days. Nevertheless, the total leaching ratio did not exceed 0.4% of the mass of the adsorbent. These results evidenced a strong bonding interaction happening between Ni and the functional groups on the surface of the adsorbent, which indicated a chemical reaction occurred in the sorption process and the reaction process can be interpreted by the following Equation:(12)$$-\mathrm{Si}-\mathrm{Ti}-P-\mathrm{Zr}-O-+N_{i}^{2+}=-\mathrm{Si}-\mathrm{Ti}-P-\mathrm{Zr}-O-\mathrm{Ni}-$$


(13)
$$-\mathrm{Si}-\mathrm{Ti}-P-\mathrm{Zr}-O-\mathrm{Ni}-+H^{+}=-\mathrm{Si}-\mathrm{Ti}-P-\mathrm{Zr}-O-H+\mathrm{Ni}-$$


To make sure the low leaching ratio of Ni ions was attributed to the chemical reaction between Ni ions and the hydroxy groups rather than the dissolving of the adsorbent in strong-acids solution, the stability property measurement of the adsorbent was also studied. Similar to the conditions in the leaching experiment, 0.1 g of the adsorbent was added into 50 mL of 0.1 M HCl solution at room temperature and the contact time was 14 days. Zirconium element was chosen, as the representative ions of the adsorbent and the concentration of Zr in the solution was measured by GFAAS. The GFAAS results of Zr ions showed that there were hardly any Zr ions in the solution, which indicated that the adsorbent was stable in a solution of strong acids and ensured that the leaching of Ni ions from the adsorbed adsorbent really originated from the reaction between Ni ions and hydrogen ions.

**Figure 11 nanomaterials-11-02314-f011:**
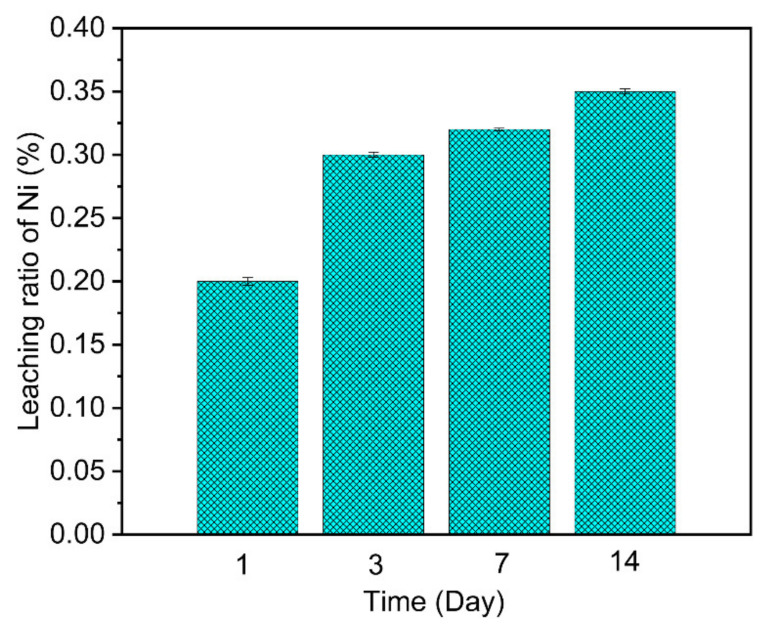
Leachability of Sr from sintered adsorbed Si-Ti-P-Zr adsorbent.

## 4. Conclusions

This study provided an excellent polymetallic phosphate adsorbent for effectively removing Ni ions from wastewater. The layer-by-layer method was firstly adopted to immobilize a useful metal phosphate adsorbent on the silica substrate which could not only significantly reduce the aggregation of the small particles, but also greatly enhance the hydraulic performance, mechanical endurance, and the abrasion resistance of the adsorbent. The Ni ions leaching and the stability of the adsorbent test indicated that the adsorbent synthesized in this paper possessed a stable ability for immobilizing Ni ions and a stable structure in the extreme acid condition, in which the total leaching ratio did not exceed 0.4% of the mass of the adsorbent within 14 days. The bath experiments revealed a fast (30 min) and high adsorption amount (50 mg/g) of Ni ions from wastewater. Considering the affinity of the adsorbent to Ni, the solution with a higher concentration of K, Mg, and Ca was used to test the affinity. The results evidenced that the composite processed higher affinity properties to Ni and the affinity order was Ni^2+^ > Ca^2+^ > Mg^2+^ > K^+^. Besides, the introduction of the thermodynamic simulations of the hydrolyzed species of Ni ions in this study provided a better way to understand the sorption mechanism of Ni ions on the adsorbent due to that this method clearly exhibiting the different hydrolyzed species of Ni ions in various pH scopes (1–14). In conclusion, the silica-based zirconium-titanium phosphate is a promising adsorbent for cleaning wastewater-containing diluted concentration of Ni ions.

## Figures and Tables

**Figure 1 nanomaterials-11-02314-f001:**
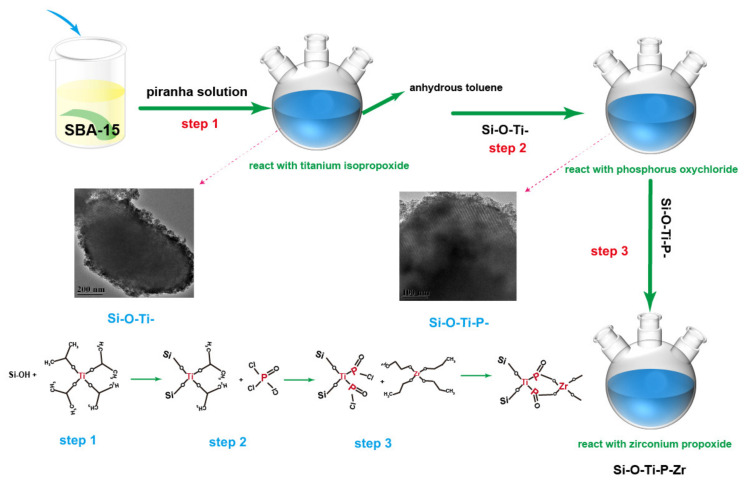
Synthesis process of the adsorbent.

**Figure 2 nanomaterials-11-02314-f002:**
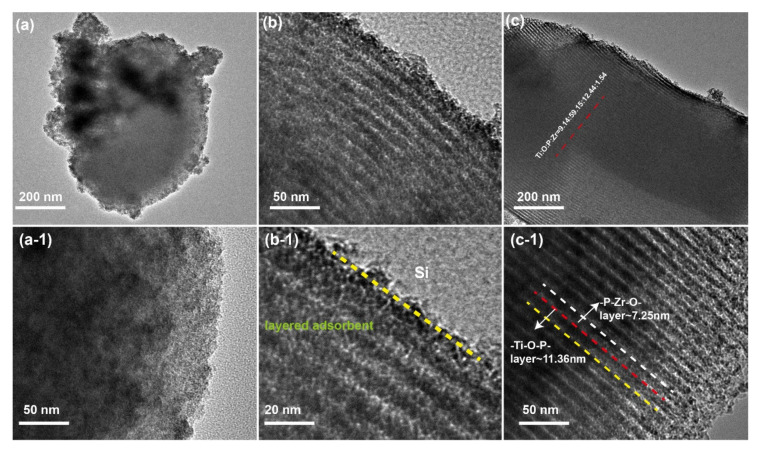
TEM micrograph of (**a**) the SBA-15 and (**a-1**) magnified image of SBA-15, (**b**) cross-section image of SiO_2_-Ti-P-Zr adsorbent, (**b-1**) magnified image of cross-section image, (**c**) the SiO_2_-Ti-P-Zr adsorbent, and (**c-1**) the magnified image of SiO_2_-Ti-P-Zr adsorbent.

**Figure 3 nanomaterials-11-02314-f003:**
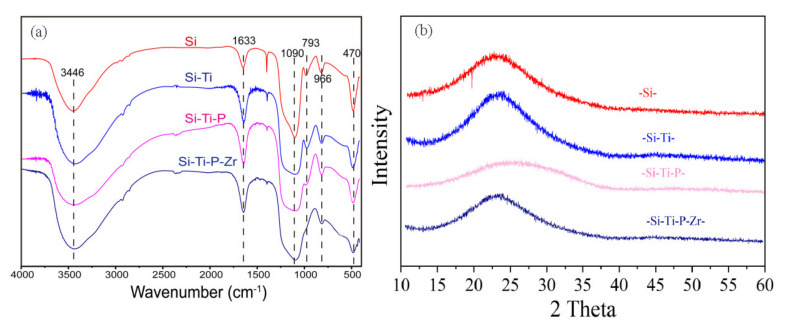
The (**a**) FT-IR and (**b**) XRD analysis of the formation process of the adsorbent (SBA-15, Si-Ti, Si-Ti-P, and Si-Ti-P-Zr).

**Figure 4 nanomaterials-11-02314-f004:**
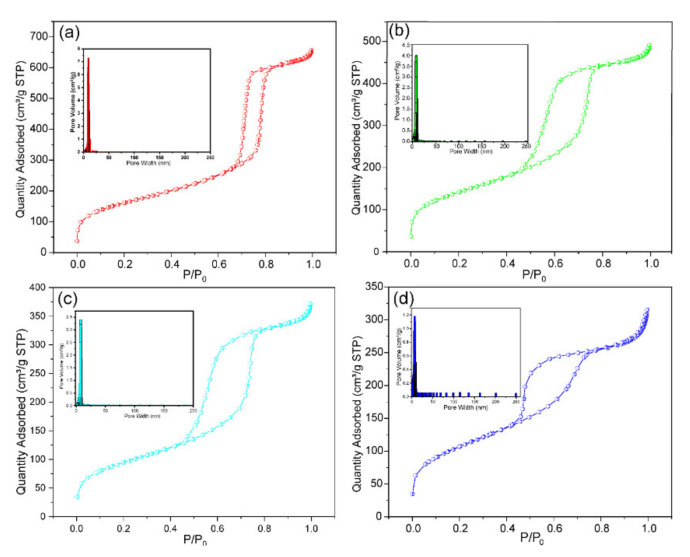
The nitrogen adsorption-desorption isotherm and the pore volume distribution of the samples: (**a**) SBA-15, (**b**) Si-Ti, (**c**) Si-Ti-P, (**d**) Si-Ti-P-Zr.

**Figure 5 nanomaterials-11-02314-f005:**
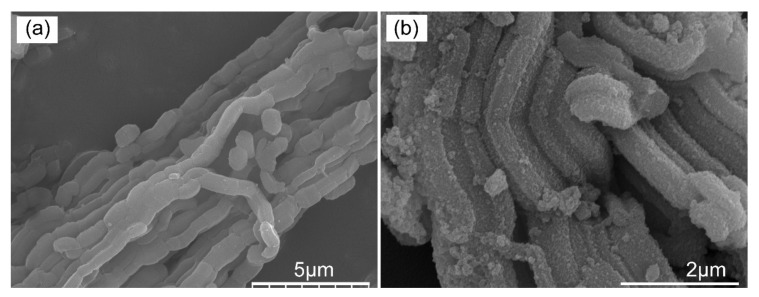
(**a**) SEM images of the pristine adsorbent and (**b**) adsorbed adsorbent.

**Figure 6 nanomaterials-11-02314-f006:**
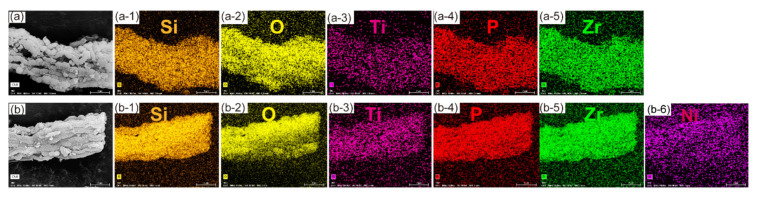
SEM images and EDS element-mappings of (**a**) the pristine adsorbent, (**a-1**) Si element distribution, (**a-2**) O element distribution, (**a-3**) Ti element distribution, (**a-4**) P element distribution, (**a-5**) Zr element distribution and (**b**) the adsorbed adsorbent, (**b-1**) Si element distribution, (**b-2**) O element distribution, (**b-3**) Ti element distribution, (**b-4**) P element distribution, (**b-5**) Zr element distribution, (**b-6**) Ni element distribution.

**Figure 7 nanomaterials-11-02314-f007:**
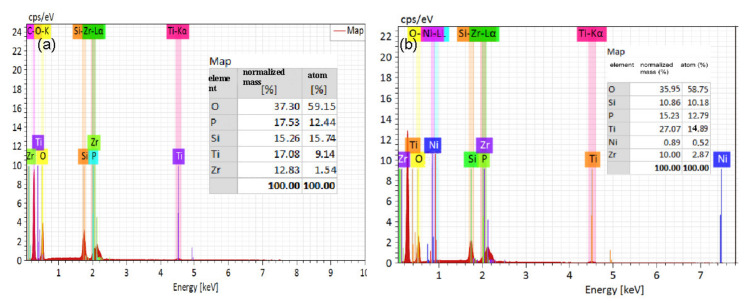
The atom percent of the elements (**a**) O, P, Si, Ti, Zr and (**b**) Si, O, Ti, P, Ni, Zr.

**Figure 8 nanomaterials-11-02314-f008:**
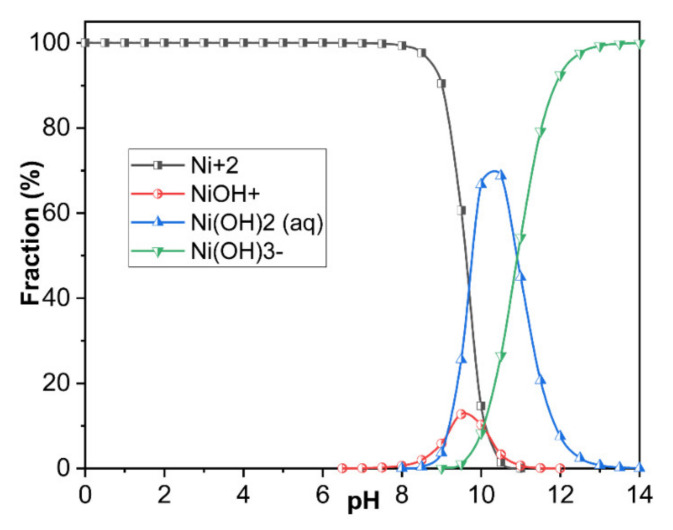
The results of the thermodynamic simulations of the hydrolyzed species of Ni ions in the pH range from 1 to 14.

**Figure 9 nanomaterials-11-02314-f009:**
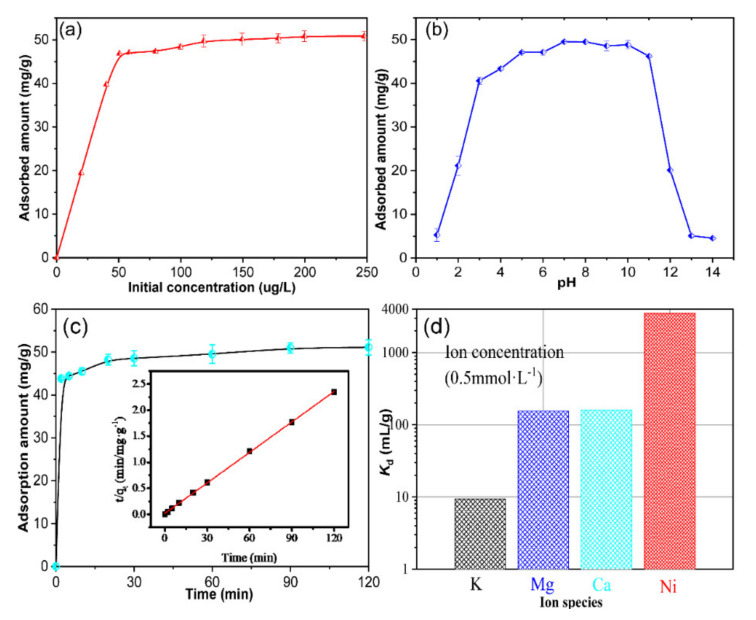
The bath experiment results. (**a**) The influence of the initial concentration on the adsorption of Ni ions; (**b**) The effect of pH on the adsorption of Ni ions; (**c**) The influence of the contact time on the adsorption of Ni ions; (**d**) The selective adsorption property of Ni from the multi-component aqueoussolution by the adsorbent.

**Figure 10 nanomaterials-11-02314-f010:**
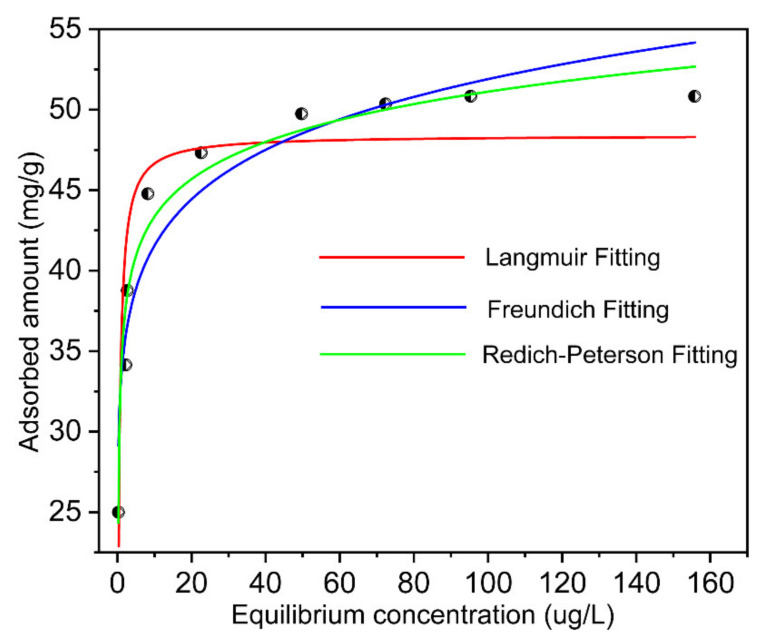
The nonlinear fitting results by Langmuir, Freundlich, and Redlich-Peterson isotherms sorption models.

**Table 1 nanomaterials-11-02314-t001:** BET surface area, pore volume, and average pore diameter by BJH adsorption isotherms.

Sample	Si Substrate	(Er.)	Si-Ti	(Er.)	Si-Ti-P	(Er.)	Si-Ti-P-Zr	(Er.)
BET Specific Surface Area (m^2^/g)	566.991	0.011	502.598	0.401	382.295	0.295	337.881	0.119
Pore Volume (cm^3^/g)	0.903	0.003	0.679	0.021	0.434	0.003	0.506	0.004
Average Pore Width (Å)	70.787	0.213	59.702	0.298	49.650	0.350	66.997	0.287

**Table 2 nanomaterials-11-02314-t002:** Ion-exchange adsorption constants obtained by fitting isotherm data with different Langmuir, Freundlich, and Redlich-Peterson models.

Isotherm Model	Estimated Isotherm Parameters			
Langmuir	*K_L_* 2.7	*q_m_* (mg/g) 48.4	*R*^2^ 0.82	
Freundlich	*K_F_* 33.3	*n* 10.3	*R*^2^ 0.91	
Redlich-Peterson	*A* 327.3	*B* 8.8	*g* 0.93	*R*^2^ 0.96

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
