# Peer review of "Study on Adsorption Behavior of Nickel Ions Using Silica-Based Sandwich Layered Zirconium-Titanium Phosphate Prepared by Layer-by-Layer Grafting Method"

_nanomaterials, 2021, doi:10.3390/nano11092314_

Round 1

Reviewer 1 Report

The manuscript requires several corrections.

  1. Introduction part should contain more references (e.g. DOI: 10.1016/j.synthmet.2011.02.022; DOI: 10.1016/j.electacta.2008.11.030; DOI: 10.1016/j.ceramint.2014.12.023; DOI: doi.org/10.1016/j.jssc.2010.07.028).
  2. Introduction part: authors should outline what is new and innovative in this work.
  3. Line 156: µg or ug?
  4. Equation 7: Ni(OH)1-

Reviewer 2 Report

This work aims to investigate fundamental properties of polymetallic phosphate adsorbent toward removing Ni ions from wastewater. The zirconium-titanium phosphate coatings were deposited via layer-by-layer grafting method. In general, this work targets important environmental goal of cleaning wastewater from heavy-metal ions and this work will be of interest to a broad scientific community. In my view several important clarification and improvements are still needed:

  • More TEM images (Fig.2) should be provided to demonstrate repeatability and give a better idea about the material.
  • EDS analysis for Fig.2 was mentioned but no data image is shown.
  • The discussion of 607 cm-1 to confirm existence of established Zr-O bands needs further experimental verification. From the provided Fig.3 (a) curves it is very difficult to draw such a conclusion.
  • Did authors conduct the FR-IR measurements after absorbing Ni ions?
  • In table 1 there are no error bars. Were the measurements repeated on several fresh systems or were conducted just once?
  • In figure 6 it is not only the SEM images but also EDX images, which needs to be clarified in the Figure caption. Have authors made more EDX studies to monitor Ni adsorption?
  • Could you please clarify, the pH studies were only carried out using the simulations or also compared with the actual measurements? In case the pH measurements were carried out you should provide the concentration of acid and base solutions used to optimize pH.

Minor points:

  • English grammar is ok, but the text sentence formulation is often difficult to understand. Also, “detailedly” researched is uncommon wording and actually, “neat” morphology can be sometimes confusing, “necessary” metal element, etc.
  • There are many abbreviations used already in the abstract, e.g. FESEM, EDS, etc. which were not yet explained. What is SBA-15 already in introduction? I am not sure everyone is familiar with it.
  • In the materials you write you used water. What water was used? Is it MilliQ or double distilled? What was the purity?
  • 4 the inset images are not readable. Fig.2 the scale bars in the inset images are not readable.

Round 2

Reviewer 1 Report

The manuscript can be published in the present form.

Author Response

Thank Reviewer again for the kind suggessions.

Reviewer 2 Report

Manuscript has been accordingly improved. It would still be good to add that the measurements in Table 1 were conducted for three fresh samples and to provide the error bar in the table or the other two measurement sets in supplementary. ” Response 2: In this work, the sample for BET surface area, pore volume and average pore diameter measurements was from three fresh systems and the results were very close and we took one of the sets of data to make table 1.”

Author Response

This manuscript is a resubmission of an earlier submission. The following is a list of the peer review reports and author responses from that submission.

Round 1

Reviewer 1 Report

The submitted manuscript is follows a typical approach how to modify porous materials to improve the adsorption of metal ions.

Although, the porous silica structure was modified by layer-by-layer approach, no detailed analysis of the formed surface near region is provided. The shown FE-SEM/EDX data and the FTIR data are not sufficiently specific to reveal the surface chemistry inside the pores.

TEM and XPS analysis could provide information on the composition and nanostructure of the surface after each modification step.

The description of the experimental methods is insufficient (e.g. no experimental data for the FTIR analysis is provided (reflection mode, energy resolution, …); not even the spectrometer model is mentioned by the authors.

TEM measurements are indicated in the experimental part, however, they are not found in the main manuscript.

Overall, I think that the manuscript is not acceptable for publication in the current state.

Reviewer 2 Report

The manuscript entitled “Study on adsorption behavior of nickel ions using silica-based sandwich layered zirconium-titanium phosphate prepared by layer-by-layer grafting method” can be published after revision. The following issues must be addressed:

  1. Introduction part must be improved in order to include more references related to this study (i.e. DOI: 10.1016/j.ceramint.2021.03.224; DOI: 10.1016/j.chemosphere.2021.131341; DOI: 10.1016/j.physb.2020.412797, DOI: 10.1016/j.cej.2021.131117; doi.org/10.1016/j.cej.2021.131117; DOI: 10.1016/j.cej.2021.131183; DOI: 10.1016/j.cej.2021.131250);
  2. Experimental: please provide the purity of each substance;
  3. Lines 256-258: Therefore, the modified adsorbent may be suitable for the adsorption of ions due to increasing surface area, and growth of crystal size. – the remark about the influence of crystal size is not clear. Provide more details.
  4. Lines 291-292 “was corresponded to the conclusions  in Fig.7.” – please reformulate.
  5. Lines 304-305 “the valence state and hydration radius of metal ions in aqueous solution were the main effect factors on the selectivity toward metal ions.” – please explain in more details.
  6. Conclusion part must contain values from the experimental results.